



# Size dependent hygroscopicity of levoglucosan and D-glucose aerosol nanoparticles

**Ting Lei[1,2], Hang Su[2,3], Nan Ma[4], Ulrich Pöschl[2], Alfred Wiedensohler[5], Yafang Cheng[1]**

[1]Minerva Research Group, Max Planck Institute for Chemistry, 55128 Mainz, Germany

[2]Multiphase Chemistry Department, Max Planck Institute for Chemistry, 55128 Mainz, Germany

[3]State Environmental Protection Key Laboratory of Formation and Prevention of Urban Air Pollution Complex, Shanghai Academy of Environmental Sciences, Shanghai 200233, China

[4]Institute for Environmental and Climate Research, Jinan University, 511443 Guangzhou, China

[5]Leibniz Institute for Tropospheric Research, 04318 Leipzig, Germany

*Correspondence to*: Yafang Cheng (yafang.cheng@mpic.de)

**Abstract:** The interaction between water vapor and aerosol nanoparticles is of great significance in atmospheric processes. However, current knowledge of hygroscopicity of sub-10 nm organic nanoparticles and their concentration-dependent thermodynamic properties (e.g., water activity) in the highly supersaturated concentration range is scarcely available. In this study, we investigate the size dependence of hygroscopicity of organics (i.e., levoglucosan, D-glucose) in size down to 6 nm using a nano-hygroscopicity tandem differential mobility analyzer (nano-HTDMA). There is a weak size dependence of the hygroscopic growth factor observed for levoglucosan and D-glucose nanoparticles with diameters down to 20 nm. However, a clear size-dependent hygroscopic growth factor is observed for D-glucose nanoparticles down to 6 nm in size. A reduction in diameters of sub-20 nm levoglucosan is observed at the dry RHs, which is explained by partial levoglucosan evaporation into gas phase, indicting high impact of volatility of sub-20 nm levoglucosan aerosol



nanoparticles. However, this also means that the hygroscopic growth factors of levoglucosan
nanoparticles with diameters below 20 nm are not possible to be determined. The use of water
activity parameterization models proposed by Kreidenweis et al. (2005) (KD, Köhler), the Extend-
Aerosol Inorganic Model (E-AIM (standard UNIFAC), and Differential Köhler Analysis (DKA)
method is to determine thermodynamic properties (e.g., water activity) of levoglucosan and D-
glucose nanodroplets as a function of solute concentration, respectively. Predicated water activity
for these aqueous organic solutions (i.e., levoglucosan, D-glucose) from the different methods are
similar to observations from references in the low solute concentration (< 20 mol kg$^{-1}$), while a
quite difference is found in the high solute concentration (> 20 mol kg$^{-1}$). In addition, we compare
hygroscopicity measurements for levoglucosan and D-glucose nanoparticles with the E-AIM
(standard UNIFAC), the ideal solution theory, and DKA predictions, respectively. The ideal
solution theory describes well the measured hygroscopic growth factors of levoglucosan with
diameters down to 20 nm and D-glucose nanoparticles with diameters higher than 60 nm,
respectively, while the E-AIM (standard UNIFAC) model can successfully predict the growth
factors of levoglucosan with diameters from 100 down to 6 nm at RH above 88-40 % (e.g., at RH
above 88 % for 100 nm D-glucose, at RH above 40 % for 6 nm D-glucose). The use of the DKA
method leads to a good agreement with measured hygroscopic growth factors of D-glucose aerosol
nanoparticles with diameters from 100 down to 6 nm.

## 43     1 Introduction

Organic aerosol nanoparticles play an important role in new particle formation, subsequent
condensation and coagulation growth, cloud condensation nuclei (CCN), and thus in affecting
visibility degradation, radiative forcing, and climate (Chylek and Coakley, 1974; Charlson et al.,



1992; Dusek et al., 2010; Cheng et al., 2012; Zhang et al., 2012; Kulmala et al., 2013). Both growth
of nanoparticles and their ability to act as CCN are directly related to its hygroscopicity that
describes the interaction between organic nanoparticles and water vapor (Köhler, 1936;
Kreidenweis et al., 2005; Su et al., 2010; Cheng et al., 2015; Wang et al., 2015). However, current
knowledge of hygroscopicity of sub-10 nm organic nanoparticles and their concentration-
dependent thermodynamic properties (e.g., water activity) in the highly supersaturated
concentration range is scarcely available.
Levoglucosan aerosol nanoparticles have attracted increasing interest in recent years (Simoneit et
al., 1999; Mochida and Kawamura, 2004; Mikhailov et al., 2009; Elias et al., 2010; Lei et al., 2014,
2018; Bhattarai et al., 2019) due to relative stability and high emission factors, which are
considered as an ideal tracer for characterization and quantitation the biomass burning (Fraser and
Lakshmanan, 2000). Also, levoglucosan is typically the most abundant species in wood burning
aerosols, which contributes substantially (16.6–30.9% by mass) to the total organics in $PM_{2.5}$
(Mochida and Kawamura, 2004). D-glucose, a hydrolysis product of cellulose and levoglucosan,
is a major pyrolysis product of wood (Mochida and Kawamura, 2004). Levoglucosan and D-
glucose substances may be representative in reproducing the hygroscopic behavior of the real
biomass burning aerosol particles (Bhandari and Bareyre. 2003; Mochida and Kawamura, 2004;
Chan et al., 2005; Koehler et al., 2006; Peng et al., 2010). Most of the previous lab studies have
been focused on investigation of the hygroscopic behavior of 100-nm levoglucosan and D-glucose
aerosol nanoparticles, which mainly utilized the humidified tandem differential mobility analyzers
(DMAs) (Mikhailov et al., 2004; Mochida and Kawamura. 2004; Koehler et al., 2006; Lei et al.,
2014; 2018). For example, Mochida and Kawamura (2004) observed that 100-nm levoglucosan
and D-glucose aerosol nanoparticles uptake/release water continuously in both deliquescence and





efflorescence modes, respectively. To our knowledge, there are no phase transitions for these
organic aerosol nanoparticles in both hydration and dehydration processes.
Early studies showed that the hygroscopicity and solubility of inorganic aerosols, such as
ammonium sulfate (AS) and sodium chloride (NaCl), exhibited a strong size dependence (Cheng
et al., 2015). Firstly, hygroscopic diameter growth factors of AS, NaCl as well as $Na_2SO_4$
nanoparticles are found to decrease with size decreases in both deliquescence and efflorescence
modes (Biskos et al., 2006a, b, c, Lei et al., 2020). Secondly, there is no significant difference in
the deliquescence relative humidity (DRH) and the efflorescence relative humidity (ERH) between
AS nanoparticles with dry diameters of 6 and 60 nm (Biskos et al., 2006b; Lei et al., 2020), while
a pronounced size dependence of the DRH of NaCl is up to 10 % RH between dry diameters of 6
and 60 nm (Biskos et al., 2006a). The behaviors of change of phase transition RH and
concentrations of $Na_2SO_4$ are between NaCl and AS (Lei et al., 2020). However, there are very few
lab studies on investigating hygroscopicity ($g_f$, DRH, ERH) of organic aerosol nanoparticles in sub-
10 nm size range (Wang et al., 2017). It is not clear how the size effect is going to influence the
hygroscopic growth of organics, especially with no DRH and ERH. Besides technique limitation
(Lei et al., 2020; Wang et al., 2017), another reason is the high diffusion of sub-100 nm organics
nanoparticles, especially in the sub-10 nm size range, which results in nanoparticle losses in the
HTDMA system (Seinfeld and Pandis, 2006).
For inorganic aerosols, the lack of thermodynamic properties of the highly supersaturated aqueous
solution nanodroplets (Tang and Munkelwitz, 1994; Tang 1996; Pruppacher and Klett, 1997; Clegg
et al., 1998) are limiting predictability of aerosol hygroscopic behavior of sub-10 nm aerosol
nanoparticles (Cheng et al., 2015). Also, there are very few thermodynamic data in the highly
supersaturated concentration for organic solution, such as levoglucosan and D-glucose (Bhandari





and Bareyre. 2003; Chan et al., 2005; Koehler et al., 2006; Peng et al., 2010). By measuring the
hygroscopic growth factor of particles of different sizes, we may be able to retrieve these
thermodynamic data using a Differential Köhler Analysis (DKA) method (Cheng et al., 2015). This
will further help us to understand the new particle formation, transportation, and their interactions
between water molecules.
In this study, we investigate the hygroscopic growth factors of levoglucosan and D-glucose
nanoparticles in size down to 6 nm using a nano-hygroscopic tandem differential mobility analyzer
(nano-HTDMA, Lei et al., 2020). Moreover, we compare our measurement data with model
prediction from the Extended Aerosol Inorganic Model (E-AIM (standard UNIFAC)) (Clegg et al.,
2001; Clegg and Seinfeld, 2006; available online: http://www.aim.env.ac.uk/aim/aim.php), the
ideal solution theory, and DKA. In addition, the use of the DKA method is to calculate
thermodynamic properties (e.g., water activity) of D-glucose nanodroplets in the highly
supersaturated concentration range and then to compare with KD-derived data (KD=Kreidenweis),
thermodynamic property data from Köhler (Kreidenweis et al., 2005), E-AIM (standard UNIFAC)
model, and references, respectively.

**2 Methodology**
**2.1 Experimental methods**
**2.1.1 Nanoparticle generation**
An electrospray is employed to generate levoglucosan and D-glucose aerosol nanoparticles of 6, 8,
10, and 15 nm using 2, 3, 5, and 10 mM aqueous solutions with 50 % volume fraction of a 20 mM
ammonium acetate buffer solution (Chen et al., 2005; Wang et al., 2015), respectively. The





generated nanoparticles are diluted by mixing with dry and filtered $N_2$ (1 l/min) and $CO_2$ (0.1 l/min),
bringing aerosol nanoparticles to a dry RH state ($\leq$ 2% RH). Subsequently, aerosol nanoparticles
pass through a $Po^{210}$ neutralizer to reach the equilibrium charge distribution (Wiedensohler 1986).
In order to avoid blocking the 25-μm capillary tube in the electrospray with high solution
concentration, the aerosol nanoparticles with diameters of 60-100 and 20 nm are generated by an
atomizer with a 0.05 and 0.01 wt % organic solution (i.e., levoglucosan and D-glucose),
respectively. The chemical substances and their physical properties are characterized in Table S1.
These solutions are prepared with distilled and de-ionized million-Q water (resistivity of 18.2 MΩ
cm at 298.15 K). Note that, the size selected by the nano-DMA1 should be the right part of peak
diameter of the number size distribution of the generated nanoparticles, which minimizes the
influence of the multiple charged nanoparticles in hygroscopicity measurements.
**2.1.2 Nano-HTDMA setup**
Figure 1 shows a schematic of the nano-HTDMA system for investigating the hygroscopic
behavior of aerosol nanoparticles, especially in the sub-10 nm size range. The detailed description,
calibration, and validation of nano-HTDMA setup have been reported in the previous paper (Lei et
al., 2020). In brief, the polydisperse aerosol nanoparticles pass through a silica gel diffusion dryer
and a Nafion gas dryer (TROPOS Model ND.070, Length 60 cm). The dry aerosol nanoparticles
at RH below 10 % are charged by a $Kr^{85}$ bipolar charger and then enter the first nano-differential
mobility analyzer (nano-DMA1, TROPOS Model Vienna-type short DMA), where a monodisperse
distribution of nanoparticles with the desired dry diameter is selected. The monodispersed
nanoparticles subsequently are exposed to the different RH conditions, which can be set to
deliquescence mode (from low RH to high RH for measuring deliquescence) or efflorescence mode
(from the high RH to low RH for measuring efflorescence). In the deliquescence mode, the dry



aerosol nanoparticles are gradually humidified to a target RH through a Nafion humidifier (NH-1,
TROPOS Model ND.070, Length 60 cm). In the efflorescence mode, after deliquescence of aerosol
nanoparticles with RH above 97% in a Nafion humidifier (NH-2: Perma Pure Model MH-110,
Length 30 cm), the deliquesced aerosol nanoparticles are stepwise dried to a target RH in NH-1.
The number size distribution of the humidified nanoparticles is then measured by a nano-
differential mobility analyzer (nano-DMA2) at a target RH through a Nafion humidifier (NH-3,
Perma Pure Model PD-100) coupled with an ultrafine condensation particle counter (CPC, TSI,
model no. 3776). To have the uniform RH within the nano-DMA2 for the accurate determination
of hygroscopicity ($g_f$, DRH, ERH) of aerosol nanoparticles, the difference between the sheath flow
RH ($RH_s$) and the aerosol flow RH ($RH_a$) upstream of the nano-DMA2 is kept <1 %. Most
importantly, the temperature difference between inlet and outlet of the nano-DMA2 is maintained
below 0.2 °C during the measurements. In addition, the residence time (e.g., 5.4 s: between the
humidifier and the nano-DMA2; 0.07 s: deliquescence for aerosol nanoparticles) is sufficient for
water-soluble aerosol nanoparticles to equilibrate with water vapor at a given RH and to occur
solid-liquid phase transition (Kerminen 1997; Duplissy et al., 2005; Raoux et al., 2007),
respectively.
**2.2 Theory and modeling methods**
**2.2.1 Köhler theory**
The fractional ambient relative humidity ($\frac{RH}{100}$) over a spherical droplet in equilibrium with the
environment is described by Köhler equation (Köhler 1936):
$\frac{RH}{100} = a_w exp\left(\frac{4\sigma_{sol}v_w}{RTG_fD_s}\right)$                                                                     (1)





where $a_w$ is the water activity of the solution droplet, $\sigma_{sol}$ is the liquid-vapor interfacial energy of
solution droplet (also called surface tension), $v_w$ is the partial molar volume of water, $R$ is the
universal gas constant, $T$ is the temperature, $G_f$ is the diameter growth factor of aerosol particles,
and $D_s$ is the dry diameter of spherical aerosol particles. The hygroscopic growth curve ($G_f$ vs $RH$)
is estimated based on the assumptions in models or theories described in the following sections

164   (2.2.2-2.2.3).

**2.2.2 Water activity**
2.2.2.1 Köhler
The expression for water activity used in the simplified Köhler theory assumes the droplet contains
$n_w$ moles of water and $n_s$ moles of nonvolatile solute.
$a_w = \frac{n_w}{n_w + v n_s}$ (2)
$v$ is the number of ions of solute present in solution ($v$=1 for organic composition). This expression
has been applied to the diluted solution (Kreidenweis et al., 2005; Koehler et al., 2006).
2.2.2.2 KD
The following KD expression is proposed by Kreidenweis et al. (2005) (KD= Kreidenweis) is to
present the relationship between $a_w$ and $G_f$ determined in hygroscopic growth measurements:
$Gf = \left[ 1 + (a + b * a_w + c * a_w^2) \frac{a_w}{1 - a_w} \right]^{\frac{1}{3}}$ (3)
The coefficients a, b, and c for organic solution droplet in this study from Lei et al. (2014, 2018)
and Estillore et al. (2017) as shown in Table S2.
2.2.2.3 DKA





Differential Köhler analysis (DKA) proposed by Cheng et al. (2015) is theoretically based on
Köhler equation (Köhler, 1936) to determine water activity by measuring hygroscopic growth
factors of aerosol nanoparticles in different sizes.
$$a_w = \frac{s_{w1}^{\left(\frac{D_{s1}}{D_{s1}-D_{s2}}\right)}}{s_{w2}^{\left(\frac{D_{s2}}{D_{s1}-D_{s2}}\right)}}$$ (4)
where $A = \frac{4v_w}{RTg_f}$,  $s_{w1}$ and $s_{w2}$ are water saturation ratio measured at the same $g_f$ but at the
different initial dry diameters $(D_{s1}, D_{s2})$, respectively. Using the DKA method can calculate the
water activity in the highly supersaturated concentration range.

**2.2.3 Growth factor**

2.2.3.1 Ideal solution growth factor
For idea solution, the hygroscopic curve can be estimated assuming that the water activity $a_w$ of
the solution containing non-volatile and non-electrolyte solute component is equal to the molar
ratio of water in the solution. Here, the partial molar volume of pure water in the solution is equal
to the molar volume of pure water. Since the hygroscopic diameter growth factor measurements
are on volume basis using nano-HTDMA system, the expression of $G_f$ as a function of molar ratio
$(x_j)$, molar mass $(M_j)$, and mass density $(\rho_j)$ of components $j$ as follows:
$$G_f = \left[\frac{\sum_j \left(x_j M_j \frac{1}{\rho_j}\right)}{\sum_{j,j \neq w} \left(x_j M_j \frac{1}{\rho_j}\right)}\right]^{\frac{1}{3}}$$ (5)
2.2.3.2 Growth factor prediction by E-AIM model
The hygroscopic growth curve of aerosol particles is commonly evaluated from Extend-Aerosol
Inorganic Model (E-AIM). It is a thermodynamic equilibrium model used for calculating phase
partitioning (gas/liquid/solid). Most importantly, the E-AIM mode can model thermodynamic



properties (e.g., water activity, liquid-vapor interfacial energy, and solution density) in the highly
supersaturated concentration solution (Dutcher et al., 2013). Also, the standard universal quasi-
chemical functional group activity coefficients (UNIFAC) within E-AIM can be used to predict
$a_w$, $\sigma_{sol}$, and $\rho_{sol}$ of organic aqueous solution (Fredenslund et al., 1975; Hansen et al., 1991). Note
that, The E-AIM calculations based on the standard UNIFAC group contribution method are to
predict hygroscopic growth factors of organic aerosol particles. (i.e.., E-AIM model (standard
UNIFAC)) growth curve as a function of RH is based on Eq. (1) and Eq. (6).).
$G_f = \left(\dfrac{\rho_s}{x_s \rho_{sol}}\right)^{\frac{1}{3}}$                    (6)
$\rho_s$ and $\rho_{sol}$ are the density of solute and solution, respectively, and $x_s$ is the solute mass fraction.
**2.2.4 Calculation of ratio of gas-phase concentration to the total concentration**
2.2.4.1 Calculation of gas-phase concentration (g/cm$^3$)
$P_A = P_A^0 \, exp\left(\dfrac{2\sigma M}{RT\rho_l R_p}\right)$                    (7)
$m_{gas} = \dfrac{PVM}{RT}$                    (8)
where $P_A$ and $P_A^0$ are vapor pressure, equilibrium vapor pressure, respectively. σ, $M$ ,
$\rho_l, and\ R_p$ mean surface tension, molecular weight of the substance, liquid-phase density, and a
droplet of radius, respectively. This equation (Eq. 8) establish a relationship between mass in gas
phase ($m_{gas}$) and pressure ($P$), volume ($V$), mole mass ($M$), the ideal gas constant ($R$), and
temperature. Here, Vapor pressure ($P$) is equal to saturated ratio of levoglucosan vapor multiplied
saturated levoglucosan vapor pressure at 293.15 K.
2.2.4.2 Calculation of total concentration of generated particles (g/cm$^3$)



$m_{total} = \frac{dN}{dlogD_p} \times dlogD_p \times \frac{\pi}{6} D_p^3 \times \rho$ (9)
where $dN$ is particle concentration, $D_p$ is the particle diameter, and $\rho$ is the density of particles.
2.2.4.3 Ratio of the gas-phase concentration to the total concentration of generated particles
$Ratio = \frac{m_{gas}}{m_{total}}$ (10)

**3 Results and discussion**
**3.1 Levoglucosan**
**3.1.1 Concentration-dependent water activity of levoglucosan solution**
Figure 2 shows KD-derived water activity of aqueous levoglucosan nanoparticles with molality up
to 140 mol kg$^{-1}$. Here, by applying a water activity parameterization model (KD, Eq. 3) to measured
growth factors of levoglucosan aerosol nanoparticles with diameters from 20 to 100 nm using a
nano-HTDMA. Chan et al. (2005) levitated single particles of ~10 μm levoglucosan at the different
RHs in an electrodynamic balance for mass measurements, and reported water activity data for
aqueous droplets with molality up to 14 mol kg$^{-1}$. These water activity data are compared with
predictions from the Köhler (Kreidenweis et al., 2005, Eq. 2) and the E-AIM model, respectively.
A good agreement between KD-derived water activity and Köhler indicates these aerosol particles
are diluted aqueous droplets with molality less than 20 mol kg$^{-1}$. However, a derivation of Köhler
from the KD-derived water activity is observed as the molality increases from 20 to 120 mol kg$^{-1}$,
indicating levoglucosan nanoparticles become the highly supersaturated. Also, a discrepancy exists
between KD-derived data and E-AIM model prediction. For DKA-derived water activity
calculations, a strong size dependence of the hygroscopic growth factors is needed for aerosol



nanoparticles in the different sizes, which is not the case for the hygroscopic measurements of
levoglucosan nanoparticles.

**3.1.2 Size dependent hygroscopicity of levoglucosan nanoparticles**

Black solid squares in Fig. 3 shows the measured humidogram of 100-nm levoglucosan
nanoparticles in both deliquescence and efflorescence modes. Levoglucosan nanoparticles uptake
water continuously from 5 % to 90 % RH. Also, they gradually release water as RH decreases
down to 5 %. The hygroscopic growth factors of levoglucosan nanoparticles in deliquescence and
efflorescence modes overlap. For example, the hygroscopic growth factors of levoglucosan
nanoparticles at 80 % RH, 87 % RH are 1.16, 1.23, respectively, in the deliquescence mode, very
close to the corresponding values in the efflorescence mode are 1.15, 1.22 (shown in Fig. S1),
suggesting that growing and shrinking of particles are in equilibrium with water vapor surrounding
moisture conditions. No prompt phase transitions of levoglucosan nanoparticles are observed in
both deliquescence and efflorescence modes. A similar non-prompt phase transition of
levoglucosan nanoparticles was observed in the previous studies (Mochida and Kawamura, 2004;
Chan et al., 2005; Svenningsson et al., 2006; Mikhailov et al., 2008; Lei et al., 2014, 2018). This
study is in good agreement with most of reference results, but there is a difference in the
hygroscopic growth factor of levoglucosan nanoparticles between Mikhailov et al. (2008) and this
study. The reason is that Mikhailov et al. (2008) used minimum mobility diameter measured in the
hydration and dehydration modes instead of the initial dry mobility diameter measured in the
hydration or dehydration modes to calculate the hygroscopic growth factor of levoglucosan
nanoparticles, which could lead to the higher hygroscopic growth factors of levoglucosan
nanoparticles than that of this study.





Figure 4 shows measured size-resolved hygroscopic growth factors of levoglucosan nanoparticles
against RH up to 90 %. There is a weak size dependence of hygroscopic growth factors of
levoglucosan nanoparticles with diameters down to 20 nm in both deliquescence and efflorescence
modes. E.g., a slight difference in hygroscopic growth factor between 100 and 20-nm levoglucosan
nanoparticles is ~0.02 at 88 % RH. In addition, E-AIM (standard UNIFAC) model and ideal
solution theory are used to predict our measurement results as shown in Fig. 4a and 4b, respectively.
E-AIM (standard UNIFAC) model is applied to estimate the hygroscopic growth of organic aerosol
nanoparticles according to UNIFAC group contribution method. Ideal solution theory is used to
describe water absorption of the ideal/diluted aqueous solution nanodroplets. Due to consideration
of Kelvin effect in model and theory, these model predictions are expected to present a size
dependence of growth factors of nanoparticles in size from 100 down to 20 nm. For example, as
shown in Fig. 4a, the thermodynamic equilibrium model (E-AIM (standard UNIFAC)) shows a
weak size dependence of the growth factors of levoglucosan nanoparticles with diameters 100, 60,
and 20 nm at low RH but a strong size dependence of growth factors at RH above 70 %. However,
the calculated growth factors of nanoparticles down to 20 nm in size are deviated from the
measured growth factors of levoglucosan nanoparticles at RH below 80 %, which is similar to the
observation of 100-nm levoglucosan hygroscopicity prediction from previous studies (Lei et al.,
2014, 2018). Lei et al. (2014, 2018) explained that the possible reason for this discrepancy is that
the E-AIM (standard UNIFAC) predictions are not suitable for organic compounds with the
strongly polar functional groups in series (Fredenslund et al., 1975; Hansen et al., 1991). Since
levoglucosan contains three OH groups in series, thus, thermodynamic properties (e.g., water
activity, surface tension) in E-AIM (standard UNIFAC) are more likely to be invalid for
levoglucosan system. However, a good agreement of growth factors of levoglucosan with





diameters 100, 60, and 20 nm is observed between measurements and predictions by ideal solution
theory as shown in Fig. 4b.
The hygroscopic growth for sub-20 nm levoglucosan nanoparticles cannot be determined with the
nano-HTDMA system because we observed significant evaporation of the dry particles in the
measurement system. Figure 5a-b shows the measured peak diameter of normalized size
distribution scanned by the nano-DMA2 and nano-DMA1 for sub-20 nm levoglucosan
nanoparticles. It is obvious that the size of nanoparticles in DMA2 is smaller than that in DMA1,
corresponding to a decrease of 22% to 50% of 15-nm and 10-nm levoglucosan nanoparticles,
respectively, indicating significant evaporation of these small levoglucosan nanoparticles in the
system. To test this hypothesis, we estimate the ratio of gas-phase concentration to the total
concentration of the generated levoglucosan nanoparticles in the different sizes. Firstly, the
calculated gas-phase concentration of levoglucosan is based on the Kelvin equation and ideal gas
equation (Eq. 7&8, Sect. 2.2.4). Figure 5c shows the vapor saturation ratio of levoglucosan as
nanodroplet diameter increases from 0 to 100 nm. The inset in Fig. 5c is an enlarged view (black
open square) of vapor saturation ratio of levoglucosan as a function of nanodroplet diameters below
20 nm. The Kelvin effect on levoglucosan nanodroplets is very weak at diameters above 20 nm,
but significantly enhanced for levoglucosan nanodroplets with diameters below 20 nm. Secondly,
the total concentration of levoglucosan particles is estimated by Eq. (9). Thus, the results of the
ratio of gas-phase concentration ($m_g$) to the total concentration ($m_t$) have been shown in Fig. 5d and
Table S3 for levoglucosan nanoparticles in the diameter range from 10 to 100 nm. It shows a slight
increase in the calculated ratio ($m_g/m_t$) for levoglucosan aerosol nanoparticles with dimeters from
100 down to 20 nm. However, the ratio of gas-phase concentration to the total concentration is
dramatically enhanced for sub-20 nm levoglucosan aerosol nanoparticles, which is consistent with



measurement observations, indicting the larger impact of evaporation of sub-20 nm levoglucosan
nanoparticles on the measurement results.

**3.2 D-glucose**

**3.2.1 Concentration-dependent water activity of D-glucose solution**

Figure 6 shows the DKA-derived water activity of aqueous D-glucose nanodroplets with diameters
from 6 nm to 100 nm with molality up to 1000 mol kg$^{-1}$ (Cheng et al., 2015, Eq. 4). Here, by
comparing with KD-derived water activity, Köhler, E-AIM model, and observation from literatures
(Comesaña et al., 2001; Peng et al., 2001; Bhandari and Bareyre, 2003; Ferreira et al., 2003), a
good agreement between them is observed in the solute concentration below 20 mol kg$^{-1}$. However,
there is a disagreement between water activity results in the highly supersaturated concentration
range (> 20 mol kg$^{-1}$).

**3.2.2 Size dependent hygroscopicity of D-glucose nanoparticles**

Figure 7 shows the measured hygroscopic growth factors of 100-nm D-glucose nanoparticles as a
function of RH. No significant difference in the hygroscopic growth factor of 100-nm D-glucose
nanoparticles is found between deliquescence and efflorescence measurement modes (Fig. S2). For
example, the measured growth factors of D-glucose nanoparticles at 81 % RH, 88 % RH are 1.16,
1.25 in the deliquescence mode, respectively, in good agreement with results in the efflorescence
mode ($g_f$=1.17 at 81 % RH, $g_f$=1.26 at 88 % RH shown in Fig. S2). Also, measured hygroscopic
growth factors of 100-nm D-glucose are consistent with results from previous studies (Mochida
and Kawamura. 2004; Suda and Petters, 2013; Estillore et al., 2017; Mikhailov and Vlasenko,
2020). No prompt phase transitions are observed during in both deliquescence and efflorescence
measurement modes. Estillore et al. (2017) observed a slightly amorphous structure of D-glucose



particles under ambient conditions using an atomic force microscopy and D-glucose particles grow
through gradual water uptake where the solid-liquid phase transition is non-discrete. Thus, a
continuous growth/shrink of diameter in both deliquescence and efflorescence modes is explained
by the lack of crystallization of D-glucose nanoparticles upon drying to low RH below 10%.
Figure 8a shows the size dependence of measured hygroscopic growth factors of D-glucose
nanoparticles in the size range from 6 to 100 nm, with differences in growth factor up to 0.14
between 100-nm and 6-nm nanoparticles at 90 % RH (Fig. S2). A weak size dependence on the
hygroscopic growth factors of D-glucose nanoparticles is observed in the size range from 20 to 100
nm, which is similar to observation for levoglucosan nanoparticles with diameters down to 20 nm.
However, there is a strong size-dependent growth factor of D-glucose nanoparticles with diameters
from 6 to 20 nm, especially at high RH, i.e., RH > ~80%. There is no evident difference in
hygroscopic growth factors of D-glucose nanoparticles at RH below 80 % in size range from 6 to
100 nm. To have a clear observation for size dependence of the hygroscopic growth factor of D-
glucose aerosol nanoparticles with diameters down to 6 nm, Fig. 8b shows the change in the
hygroscopic growth factor of D-glucose aerosol nanoparticles with diameters from 100 down to 6
nm at 87 % RH. The hygroscopic growth factor of D-glucose nanoparticles is almost unchanged
with diameters from 20 to 100 nm. However, a markedly increase in the hygroscopic growth factor
of D-glucose aerosol nanoparticles is observed as size increases from 6 to 20 nm. E-AIM model
predict well the measured hygroscopic growth factors of D-glucose with diameters smaller than 15
nm at 87 % RH, while ideal solution theory agrees with hygroscopic measurement results of D-
glucose with diameters higher than 60 nm at the same RH. The use of DKA methods leads to a
good agreement between measurements and model predictions.



The measured hygroscopic growth factors of D-glucose nanoparticles with diameters of 6 and 100
nm are compared with the model and theory shown in Fig. 9, Fig. S3, and Fig. S4, respectively.
Note that, E-AIM (standard UNIFAC) model prediction is optimized for organic compounds with
lesser polar groups in series, i.e., intramolecular interaction, such as hydrogen bond between polar
groups, may result in model prediction inaccuracy. Ideal solution theory is applied to predict the
hygroscopic growth factor of organics in the ideal solution. Figure 9a and Fig. S3 show that the
measured growth factors of 100-nm D-glucose nanoparticles are lower than predicted growth
factors from E-AIM (standard UNIFAC) model, especially at RH below 85 %. Also, E-AIM
(standard UNIFAC) model could predict well the measured hygroscopic growth factor of 6-nm D-
glucose aerosol nanoparticles at RH above 40 % shown in Fig. 9a and Fig. S3. The possible reason
for discrepancies between E-AIM (standard UNIFAC) model and measurements is inaccurate
thermodynamic parameters (e.g., water activity, surface tension) estimated by the E-AIM (standard
UNIFAC) model without consideration intramolecular interaction. D-glucose contains five OH
groups in series, hydrogen bond could potentially exist and affects the E-AIM (standard UNIFAC)
model-measurement agreement for D-glucose aerosol nanoparticles system. Using ideal solution
theory is to predict the hygroscopic curve of D-glucose nanoparticles with diameters of 6-100 nm
shown in Fig. 9b and Fig. S3. There is a good agreement between measured growth factors of 100-
nm D-glucose and ideal theory predictions. This suggests that thermodynamic parameters (e.g.,
water activity, surface tension, and solution density) assumed by the ideal solution theory are
accurate to use in Eq. (1) and (2) for predicting the hygroscopic curve of D-glucose nanoparticles
with large sizes (e.g., 60, 100 nm). However, an underestimation of growth factors of 6-nm D-
glucose nanoparticles has been shown in Fig. 9b and Fig. S3 by ideal solution theory prediction at
RH above 30 %. The possible reason is the unfavorable assumption of ideal solution theory. As D-
glucose size decreases from 20 to 6 nm, D-glucose nanodroplets could be highly supersaturated in





concentration compared to the dilution solution. However, the current thermodynamic models (e,g.,
E-AIM) mostly rely on the concentration-dependent thermodynamic properties (such as water
activity) derived from the measurements of large aerosol particles or even bulk samples (Tang and
Munkelwitz, 1994; Tang, 1996; Pruppacher and Klett, 1997; Clegg et al., 1998). They are thus
difficult or impossible to apply to describe the hygroscopic behavior of sub-10 nm nanoparticles,
which can often be supersaturated in concentration compared to bulk solutions (Cheng et al., 2015;
Wang et al., 2018). Thus, nanosize effect on these thermodynamic properties have been taken into
account in the models and theories (Cheng et al., 2015). Combination of DKA methods and
hygroscopic measurements of aerosol nanoparticles in the different sizes can use to determine the
thermodynamic properties (e.g., water activity) in the highly supersaturated concentration range
(Cheng et al., 2015). Therefore, as shown in Fig. 9c and Fig. S4, the use of the DKA method leads
a good agreement with the measured hygroscopic growth factors of Glucose nanoparticles with
diameters from 100 down to 6 nm.

**4 Conclusions**
In this study, we investigate the hygroscopic behavior of levoglucosan and D-glucose nanoparticles
with diameters down to 6 nm using a nano-HTDMA. Due to the larger impact of evaporation of
sub-20 nm levoglucosan nanoparticles in the nano-HTDMA system, we measure hygroscopic
growth factor of levoglucosan with diameters down to 20 nm. There is a weak size dependence of
hygroscopic growth factor of levoglucosan and D-glucose with diameters down to 20 nm, while a
strong size dependence of the hygroscopic growth factor of D-glucose has been clearly observed
in the size range from 6 to 20 nm. No prompt phase transitions occur in both deliquescence and
efflorescence modes for both levoglucosan and D-glucose nanoparticles. By comparing with the





KD-derived water activity, Köhler, E-AIM model, and DKA-derived data, the predicted water
activity of aqueous organic solution (levoglucosan and D-glucose) is consistent with observation
data from references in the low solute concentration (< 20 mol kg$^{-1}$) but failed in the solute
concentration (> 20 mol kg$^{-1}$). In addition, ideal solution theory predicts well the hygroscopic
behavior of two specific organics with diameters higher than 60 nm (levoglucosan and D-glucose),
while hygroscopic growth factor of D-glucose down to 6 nm in size is in good agreement with E-
AIM (standard UNIFAC) model prediction at high RH. The use of the DKA method leads to a
good agreement with measured hygroscopic growth factor of glucose nanoparticles with diameters
from 100 down to 6 nm.
Biomass burning is an important source of anthropogenic atmospheric aerosols. Aerosol particles
in the biomass burning smoke enriched with hygroscopic behavior are suggested to act as efficient
CCN. It is well known that aerosol population can appear as externally mixed or internally mixed
(homogeneously internally, core-shell internally) in the biomass burning processes. The mixing
structure has an important effect on the hygroscopic behavior of aerosol particles, especially for
sub-100 nm size range. We will be able to investigate the effect of the mixing state on the
hygroscopic behavior of aerosol nanoparticles from biomass burning in different sizes. This will
further help us to understand their interaction with water vapor.

**Data availability**
Reader who are interested in the data should contact Yafang Cheng (Yafang.cheng@mpic.de).
**Competing interests**





Some authors are members of the editorial board of journal Atmospheric Chemistry Physics. The
peer-review process was guided by an independent editor, and the authors have also no other
competing interests to declare
**Acknowledgement**
This study was supported by the Max Planck Society (MPG) and Leibniz Society. T.L
acknowledges the support from China Scholarship Council (CSC). Y. C. would like to
acknowledge the Minerva Program of MPG.
**Author contributions**: Y.C. and H.S. designed and led the study. T.L. performed the experiments.
All co-authors discussed the results and commented on the manuscript.  T.L. wrote the manuscript
with input from all co-authors

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






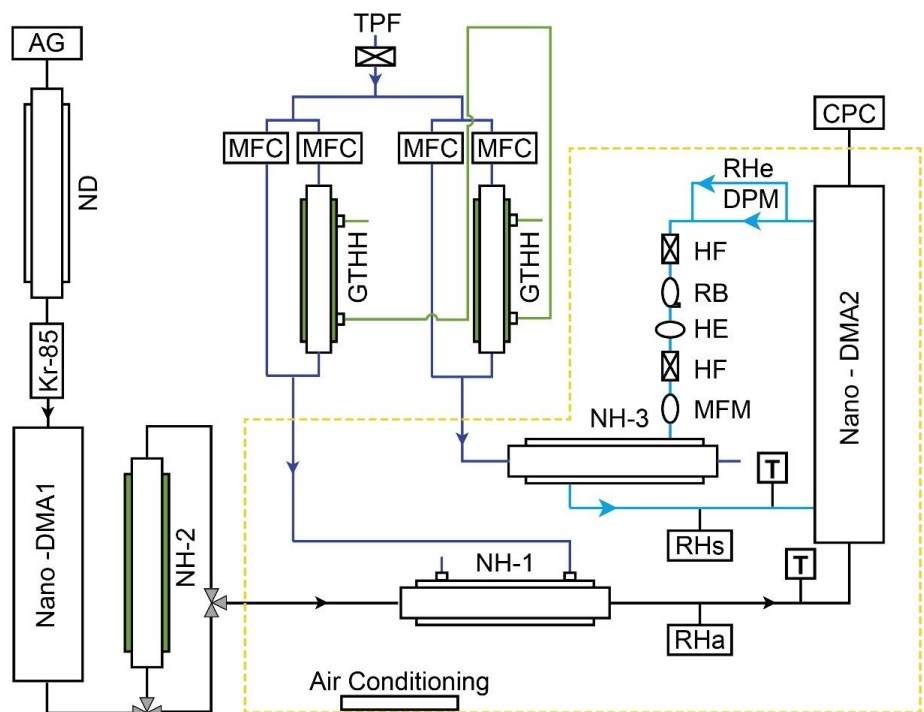

**Figure 1.** Experimental setup of the nano-HTDMA. Here, AG: aerosol generator (aerosol atomizer or electrospray);
ND: nafion dryer; Kr-85: Krypton source aerosol neutralizer; Nano-DMA: nano differential mobility analyzer; TPF:
total particle filter; HF: hydrophobic filter; MFC: mass flow controller; MFM: mass flow meter; RB: recirculation
blower; DPM: dew point mirror; GTHH: Gore-Tex humidifier and heater; NH: nafion humidifier; HE: heat exchanger;
CPC: condensation particle counter; Black line: aerosol line; Blue line: sheath line; Royal blue line: humidified air;
Green line: MilliQ water (resistivity of 18.2 MΩ cm at 298.15 K). $RH_a$ and $RH_s$ (measured by RH sensors) represent
the RH of aerosol and sheath flow in the inlet of nano-DMA2, respectively. $RH_e$ (measured by dew point) represents
the RH of excess air. T represent the temperature of aerosol and sheath flow in the inlet of nano-DMA2, respectively.








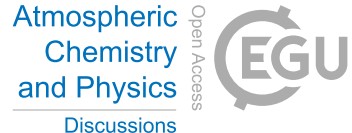


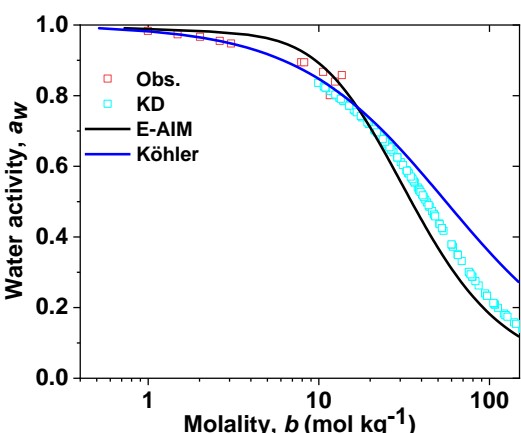

**Figure 2**. Concentration-dependent water activity ($a_w$) of levoglucosan solution. The KD-derived $a_w$ (KD=Kreidenweis,

cyan open square) is compared with observations (red open square), E-AIM (Extend-Aerosol Inorganic Model, black

line), and $a_w$ model (Köhler, blue line).

























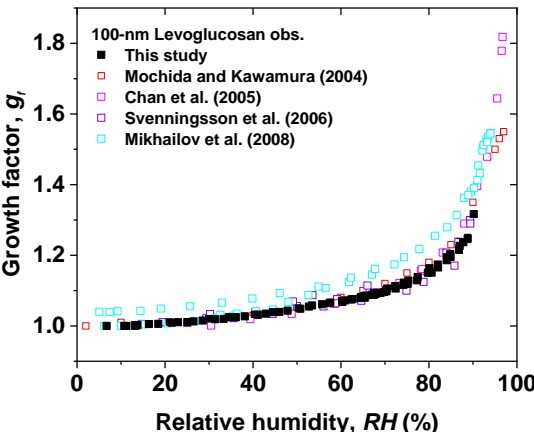


**Figure 3.** Hygroscopic diameter growth factor ($G_f$) of levoglucosan particles with dry diameter of 100 nm in both

deliquescence and efflorescence mode processes (black solid square). The measured data compared with literature data

from Mochida and Kawmura (2004) in both deliquescence and efflorescence modes (red open square), Chan et al.

(2005) in the deliquescence mode (magenta open square), Svenningsson et al. (2006) in the deliquescence mode (violet

open square), and Mikhailov et al. (2008) in both deliquescence and efflorescence modes (cyan open square).














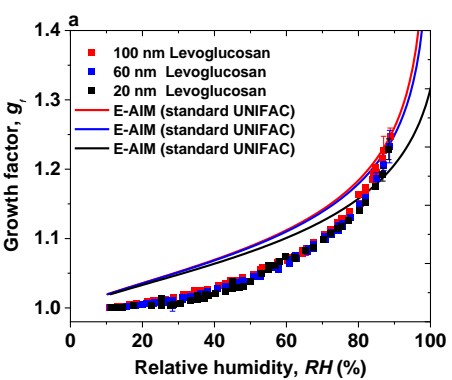 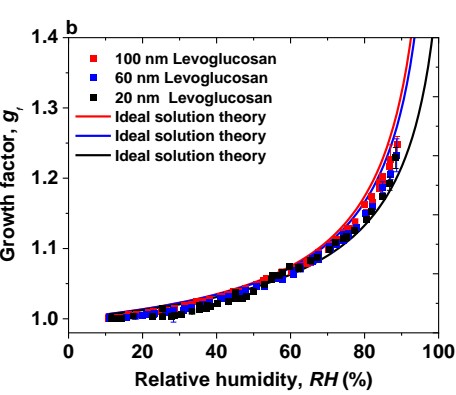

**Figure 4.** Hygroscopic diameter growth factor ($G_f$) of levoglucosan particles with dry diameter of 100 nm (red square), 60 nm (blue square), and 20nm (green square). Köhler model curves are based on: **(a)** E-AIM (standard UNIFAC) (100 nm: red, 60 nm: blue, 20 nm: green line), **(b)** ideal solution theory (100 nm: red, 60 nm: blue, 20 nm: green line).



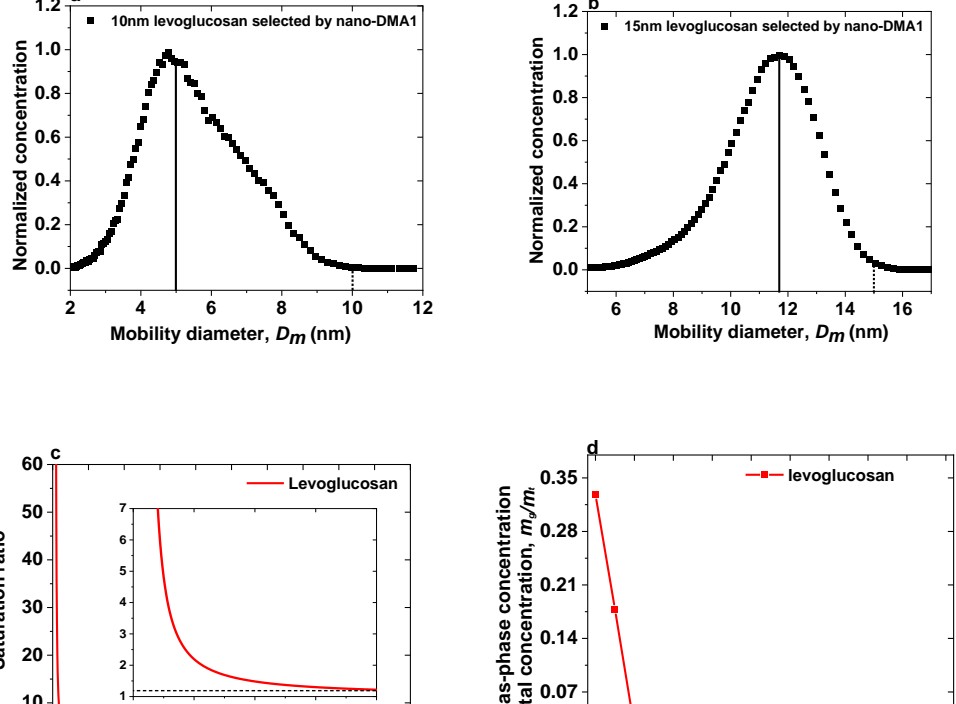



**Figure 5.** The normalized size distributions scanned by nano-DMA2 for: **(a)** 10 nm and **(b)** 15-nm levoglucosan at 10%

at 298K. The dotted lines mark the diameters of the monodispersed nanoparticles selected by the nano-DMA1. The

back solid lines mark the peak diameters from the normalized size distributions scanned by the nano-DMA2. **(c)** Vapor

saturation ratio of levoglucosan as a function of nanodroplet diameter according to the Kelvin equation. The diameter

range 0-20 nm for the saturation ratio of levoglucosan particles is shown as an inset. The value of surface tension of

pure levoglucosan is 0.0227104 [J m$^{-2}$]. **(d)** The ratio of gas-phase concentration ($m_g$) to the total concentration ($m_t$) of

levoglucosan nanoparticles against diameter.











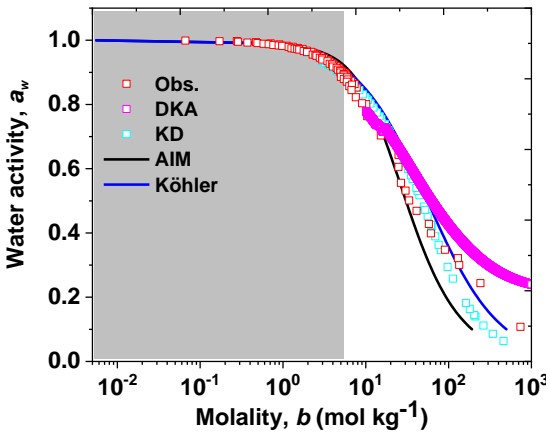

**Figure 6**. Concentration-dependent water activity ($a_w$) of levoglucosan solution. The DKA-derived $a_w$ (Differential

Köhler Analysis, magenta open square) is compared with observations (red open square), E-AIM (Extend-Aerosol

Inorganic Model, black line), $a_w$ model (Köhler, blue line), and parameterization model for $a_w$ (KD=Kreidenweis,

cyan open square). The light grey shaded areas mark the sub-saturated concentration with respect to bulk solution.











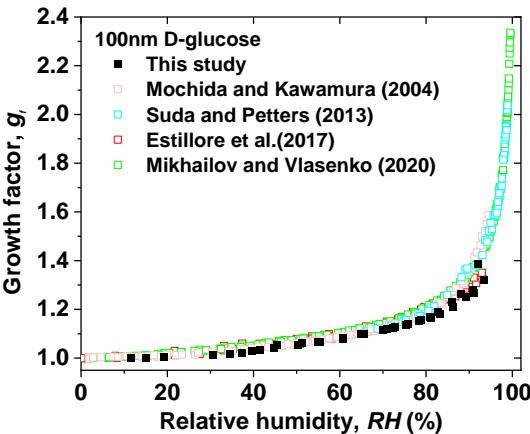


**Figure 7.** Hygroscopic diameter growth factor ($G_f$) of D-glucose particles with dry diameter of 100 nm in both

deliquescence and efflorescence modes (black solid square). The measured data compared with reference data from

Mochida and Kawmura (2004) in both deliquescence and efflorescence modes (pink open square), Suda and Petters,

(20017) in deliquescence mode (violet open square), Estillore et al., (2017) in both deliquescence and efflorescence

modes (red open square), and Mikhailov and Vlasenko (2020) in both deliquescence and efflorescence modes (green

open square).














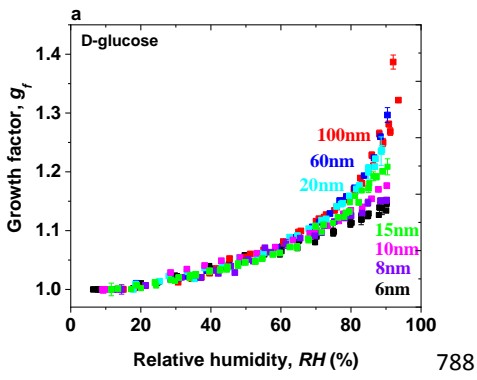
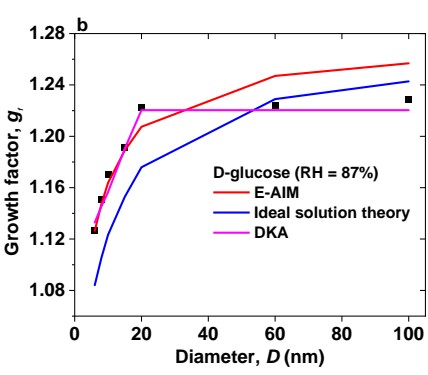


**Figure 8. (a)** Hygroscopic diameter growth factor ($G_f$) of D-glucose nanoparticles with dry diameters of 100 nm (red square), 60 nm (blue square), 20 nm (cyan square), 15 nm (green square), 10 nm (pink square), 8 nm (royal square), and 6 nm (black square). **(b)** Hygroscopic diameter growth factor ($G_f$, black square) of D-glucose nanoparticles with dry diameters from 6 to 100 nm at 87% RH. The measured hygroscopic growth factors of D-glucose nanoparticles with diameters from 100 down to are compared with E-AIM model (red line), ideal solution theory (blue line), and DKA prediction (pink line).






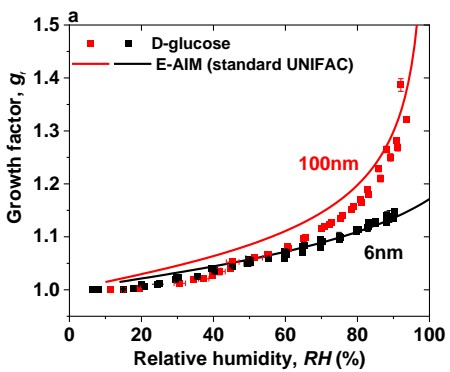
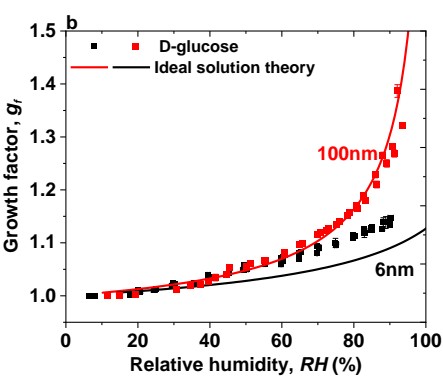


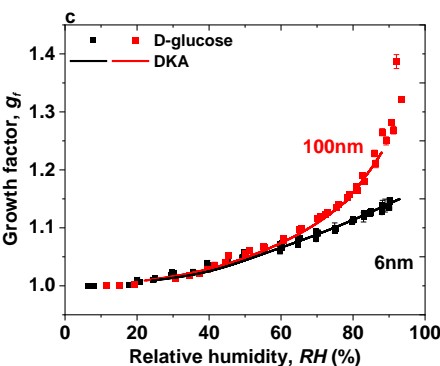

**Figure 9.** Hygroscopic diameter growth factor ($G_f$) of D-glucose nanoparticles with dry diameters of 100 nm (red
square) and 6 nm (black square). Köhler model curves are based on: **(a)** AIM (standard UNIFAC), (100 nm: red, 6 nm:
black line), **(b)** ideal solution theory (100 nm: red, 6 nm: black line), and **(c)** DKA mode (100 nm: red, 6 nm: black
line).



