# Peer review of "Size dependent hygroscopicity of levoglucosan and D-glucose aerosol nanoparticles"

_Atmospheric Chemistry and Physics, 2022_

## Author Comment (AC1)

*Response to comments by anonymous referee #1:*

*This manuscript presents hygroscopic properties of levoglucosan and D-glucose nanoparticles, mainly growth factors, through combined laboratory nano-HTDMA measurements and model predictions. Measurement data of hygroscopic properties of nanoparticles, especially for sizes down to 6 nm are scarcely lacking. The results from this study may make important contributions to improve our understanding of the hygroscopic properties of small nanoparticles and thus their important roles in new particle formation and growth, also in aerosol aging, in particular, the important roles of the two important biomass burning product in the above aerosol formation and aging processes. The following issues/concerns need to be fully resolved before the manuscript would be considered.*

**Response:** We are grateful to referee #1 for her/his comments and suggestions to improve our manuscript. We have implemented changes based on these comments in the revised manuscript in red. In the following point-by-point response, the reviewer's comments are in italic font and marked blue. The lines mentioned are with respect to the Atmospheric Chemistry and Physics Discussions (ACPD) version.

*General comments:*

*(1) Abstract: The abstract is not concise and not getting to the main ideas of the paper. L37-39, levoglucosan is mentioned in the sentence, however, the RH examples are for D-glucose, why?*

**Response:** Thank you for the comments. We have revised the abstract to be more concise and clarified the sentence in line 37-39, and now the abstract read as:

"**Abstract:** The interaction between water vapor and aerosol nanoparticles is important in atmospheric processes. Hygroscopicity of sub-10 nm organic nanoparticles and their concentration-dependent thermodynamic properties (e.g., water activity) in the highly supersaturated concentration range are, however, scarcely available. Here we investigate the size dependence of hygroscopicity of organics (i.e., levoglucosan, D-glucose) in dry particle diameter down to 6 nm using a nano-hygroscopicity tandem differential mobility analyzer (nano-HTDMA). Our results show that there is only a weak size dependent hygroscopic growth of both levoglucosan and D-glucose nanoparticles with diameters down to 20 nm. In the diameter range smaller than 20

nm (down to 6 nm), we observed a strong size-dependent hygroscopic growth for D-glucose nanoparticles. The hygroscopic growth factors cannot be determined for levoglucosan below 20 nm due to its evaporation. In addition, we compare hygroscopicity measurements for levoglucosan and D-glucose nanoparticles with the E-AIM (standard UNIFAC), the ideal solution theory, and DKA predictions, respectively. The ideal solution theory describes well the measured hygroscopic growth factors of levoglucosan with diameters down to 20 nm and D-glucose nanoparticles with diameters higher than 60 nm, respectively, while the E-AIM (standard UNIFAC) model can successfully predict the growth factors of D-glucose nanoparticles with diameters from 100 down to 6 nm at RH above 88-40 % (e.g., at RH above 88 % for 100 nm D-glucose, at RH above 40 % for 6 nm D-glucose). The use of the DKA method leads to a good agreement with measured hygroscopic growth factors of D-glucose aerosol nanoparticles with diameters from 100 down to 6 nm. Predicted water activity for these aqueous organic solutions (i.e., levoglucosan, D-glucose) from different parameterization methods agrees well with observations in the low solute concentration range ($< 20$ mol kg$^{-1}$), and start to deviate from observations in the high solute concentration ($> 20$ mol kg$^{-1}$)."

*(2) Introduction: What is the purpose for the authors to compare the availability of the thermodynamic data between highly supersaturated inorganic aqueous solutions and organic solutions such as levoglucosan and D-glucose (the paragraph in L88-97)? A clear connection and comparability between the inorganic and organic solutions is needed.*

**Response:** Thank you for the comments. The purpose of the paragraph in L88-97 is mainly to explain that the current thermodynamic models are difficult or impossible to predict the hygroscopic growth factor of sub-10 nm organic nanodroplets due to lack of thermodynamic data such as water activity and surface tension from models available in the high supersaturation concentration range. Thus, a Differential Köhler Analysis (DKA) method (Cheng et al., 2015) is introduced to retrieve these thermodynamic data of organic nanodroplets, especially in the high supersaturation concentration.

**We have revised the paragraph in L88-97 and now they read as:**

Thermodynamic model is widely used to predict the hygroscopic growth factor of organic aerosol particles as a function of RH (Bhandari and Bareyre. 2003; Chan et al., 2005; Koehler et al., 2006; Peng et al., 2010). The thermodynamic model needs thermodynamic data such as water activity, liquid-vapor interfacial energy (surface tension), and density of organic aqueous solutions (Tang and Munkelwitz, 1994; Tang 1996; Pruppacher and Klett, 1997; Clegg et al., 1998). Because

nanodroplets can become more highly supersaturated where no thermodynamic data are available,, it makes the current thermodynamic model difficult or impossible to predict the hygroscopic behavior of organic aerosol nanoparticles. Cheng et al. (2015) pointed out that size effect might be taken models into account. By measuring the hygroscopic growth factor of organic nanoparticles (e.g., levoglucosan and D-glucose) of different sizes, we may be able to retrieve these thermodynamic data using a Differential Köhler Analysis (DKA) method (Cheng et al., 2015). This will further help us to understand the new particle formation, transportation, and their interactions between water molecules.

*(3) Methodology: There are too many subsections in this part. Some of them are very short. It is not necessary to divide this part into so many subsections. Better rearrangement for this section is needed.*

**Response:** Many thanks. According to reviewer's suggestions, we have rearranged for this section "2.2 Theory and Modeling Methods". The 2.2.1-2.2.3 sections have been revised in the 2.2 section. The 2.2.4 section has been shown in SI section.

**Line 208-222, we have moved 2.2.4 section in SI section.**

**Line 154-207, we have revised 2.2.1-2.2.3 sections.**

**Related additions and changes included in the revised manuscript:**

**2.2 Theory and modeling methods**

**2.2.1 Köhler theory**

[revised manuscript text omitted]

*(4) In Fig.2, it is not clear where the observations are from, is it from this study or from Chan et al.? The authors ought to clarify the data source. If it is from this study, why to mention Chan et al.'s measurements. If it is from the latter, why a comparison between the two is not seen in the figure.*

**Response:** Thank you for your comments. Yes, observations are from Chan et al. (2005). We have revised this Fig. 2. Also, following the review's suggestions, we have revised the Fig. 6, and now they have been shown as follows:

[Figure]

**Figure 2**. Concentration-dependent water activity ($a_w$) of levoglucosan solution. The KD-derived $a_w$ (KD=Kreidenweis, cyan open square) is compared with observations (red open square), E-AIM (Extend-Aerosol Inorganic Model, black line), and $a_w$ model (SKT, blue line). The light grey shaded areas mark the sub-saturated concentration with respect to bulk solution.

[Figure]

**Figure 6**. Concentration-dependent water activity ($a_w$) of D-glucose solution. The DKA-derived $a_w$ (Differential Köhler Analysis, magenta open square) is compared with observations (red open square), E-AIM (Extend-Aerosol Inorganic Model, black line), $a_w$ model (SKT, blue line), and parameterization model for $a_w$ (KD=Kreidenweis, cyan open square). The light grey shaded areas mark the sub-saturated concentration with respect to bulk solution.

*(5) L287-309, it is quite interesting to note that the levoglucosan nanoparticles become significantly evaporated below 20 nm. The authors attribute this trend to the Kelvin effect. However, if it is because of Kelvin effect, why it does not occur for other organic nanoparticles such as D-glucose?*

**Response:** Many thanks. The impact of enhanced evaporation on the particle size depends on the volatility of organic compounds. For low volatile compounds, a much less fraction exists in the gas phase, which is still negligible given enhanced evaporation due to the Kelvin effect. However, for semi-volatile compound such as Levoglucosan (Hennigan et al., 2010), the effect is strong below 20 nm. Different from levoglucosan, the volatility of D-glucose is much smaller than that of levoglucosan, e. g., the vapor pressure of D-glucose and levoglucosan is 0.0026 Pa and 1.47 Pa at 122 °C (Oja and Suubery 1999). Therefore, there is an obvious partial levoglucosan evaporation from DMA1 to DMA2 within several seconds, but no clear evaporation for D-glucose aerosol nanoparticles.

**Related additions and changes included in the revised manuscript:**

**Line 300-301, we added and revised: "**Levoglucosan is semi-volatile at ambient condition (Hennigan et al., 2010). Due to Kelvin effect, sub-20 nm levoglucosan aerosol particles become more volatile."

**Line 309, we added: "**Therefore, there is an obvious partial levoglucosan evaporation from DMA1 to DMA2 within several seconds"

*(6) Conclusions: L408-415, it is abrupt to introduce the biomass burning and the mixing state in the conclusion. What is the point to present them as illustrated at the end of the paper?*

**Response:** Thank you for your comments. An introduction of biomass burning and the mixing state is main to extend our study in future.

We have removed this part from the conclusion.

*Minor comments:*

*(1) L23, indicting? You mean "indicating"? L29, predicating? Predictedï¼›L57ï¼Œquantification of the biomass; L64-65, have focused on.*

**Response:** Many thanks. It is "indicating" and we have corrected this. We have also carefully checked the whole manuscript, and included additional corrections of grammar, wording, and sentence structure.

**Line 21-24, we revised:** "The hygroscopic growth factors cannot be determined for levoglucosan below 20 nm due to its evaporation."

**Line 29-32, we revised:** "Predicted water activity for these aqueous organic solutions (i.e., levoglucosan, D-glucose) from different parameterization methods agrees well with observations in the low solute concentration range ($< 20$ mol kg$^{-1}$), and start to deviate from observations in the high solute concentration ($> 20$ mol kg$^{-1}$)."

**Line 54-58, we revised:** "Levoglucosan aerosol nanoparticles have attracted increasing interest in recent years (Simoneit et al., 1999; Mochida and Kawamura, 2004; Mikhailov et al., 2009; Elias et al., 2010; Lei et al., 2014, 2018; Bhattarai et al., 2019) due to relative stability and high emission factors, which are considered as an ideal tracer for characterization and quantification the biomass burning (Fraser and Lakshmanan, 2000)."

**Line 64-68, we revised:** "Most of the previous lab studies have focused on investigation of the hygroscopic behavior of 100-nm levoglucosan and D-glucose aerosol nanoparticles, which mainly utilized the humidified tandem differential mobility analyzers (DMAs) (Mikhailov et al., 2004; Mochida and Kawamura. 2004; Koehler et al., 2006; Lei et al., 2014; 2018)."

*(2) L83-83, this sentence seems quite awkward and it is suggested to change to "It is not clear how the size effect influences the hygroscopic growth of organics, especially those without DRH and ERH".*

**Response:** Many thanks. We have revised in the following sentence and now they read as:

**Line 83-84:** "It is not clear how the size effect influences the hygroscopic growth of organics, especially those without DRH and ERH."

*(3) L85, organic, not organics; L90, "are limiting" changed to "limit" ï¼›L173-174, please revise the ill sentence; L183, there is no A in eq. 4, please double check; L188, ideal, not idea; L195, Growth factor predicted by ...; L202-203, no coma is needed after note that and "are to predict" here "are to" is redundant and should be deleted; L214, the equation (equation (8)) shows the relationship between...; L216, no capital for V of vapor.*

**Response:** Thank you for your comments. We have revised in the following sentence and now they read as:

**Line 84-87:** "Besides technique limitation (Lei et al., 2020; Wang et al., 2017), another reason is the high diffusion of sub-100 nm organic nanoparticles, especially in the sub-10 nm size range, which results in nanoparticle losses in the HTDMA system (Seinfeld and Pandis, 2006)."

**Line 173-174:** "The following KD expression proposed by Kreidenweis et al. (2005) (KD= Kreidenweis) is to present the relationship between $a_w$ and $G_f$ determined in hygroscopic growth measurements:"

**Line 183-184, we deleted and revised:** "where $s_{w1}$ and $s_{w2}$ are water saturation ratio measured at the same $g_f$ but at the different initial dry diameters ($D_{s1}, D_{s2}$), respectively."

**Line 188-190:** "For ideal solution, the hygroscopic curve can be estimated assuming that the water activity $a_w$ of the solution containing non-volatile and non-electrolyte solute component is equal to the molar ratio of water in the solution."

**Line 195, we deleted:** "2.2.3.2 Growth factor prediction by E-AIM model"

**Line 202-205:** "Note that the E-AIM calculations based on the standard UNIFAC group contribution method predict hygroscopic growth factors of organic aerosol particles. (i.e.., E-AIM model (standard UNIFAC)) growth curve as a function of RH is based on Eq. (1) and Eq. (6).)."

**Line 214-216:** "This equation (Eq. 8) shows a relationship between mass in gas phase ($m_{gas}$) and pressure ($P$), volume ($V$), mole mass ($M$), the ideal gas constant ($R$), and temperature."

**Line 215:** According to the following equation, volume ($V$) is capital.

$$m_{gas} = \frac{PVM}{RT} \tag{R1}$$

*(4) L228-230, no subject is found in this sentence; L237, become highly supersaturated and article "the" should not be added before highly; L261, those of this study.*

**Response:** Thank you for your comments. We have revised in the following sentence and now they read as:

**Line 227-230, we revised:** "By applying a water activity parameterization model (KD, Eq. 3) to measured growth factors of levoglucosan aerosol nanoparticles with diameters from 20 to 100 nm using a nano-HTDMA, as shown in Fig. 2, we obtain water activity of aqueous levoglucosan nanoparticles with molality up to 140 mol kg$^{-1}$."

**Line 235-237, we revised:** "However, a derivation of SKT from the KD-derived water activity is observed as the molality increases from 20 to 120 mol kg$^{-1}$, indicating levoglucosan nanoparticles become highly supersaturated."

**Line 257-261, we revised:** "The reason is that Mikhailov et al. (2008) used minimum mobility diameter measured in the hydration and dehydration modes instead of the initial dry mobility diameter measured in the hydration or dehydration modes to calculate the hygroscopic growth

factor of levoglucosan nanoparticles, which could lead to the higher hygroscopic growth factors of levoglucosan nanoparticles than those of this study.

*(5) L243-261, are the notations between "deliquescence and efflorescence modes" and "hydration and dehydration modes" the same? Please clarify.*

**Response:** Many thanks. Yes, they are the same. We have changed the notion of hydration and dehydration mode to deliquescence and efflorescence modes to avoid confusion.

*(6) L308, indicating, not indicting; L316, among them? L328, delete during*

**Response:** Thank you for your comments. We have revised in the following sentence and now they read as:

**Line 306-309, we revised:** "However, the ratio of gas-phase concentration to the total concentration is dramatically enhanced for sub-20 nm levoglucosan aerosol nanoparticles, which is consistent with measurement observations, indicating the larger impact of evaporation of sub-20 nm levoglucosan nanoparticles on the measurement results."

**Line 315-316, we revised:** "a good agreement among them is observed in the solute concentration below 20 mol kg$^{-1}$."

**Line 328-329, we revised:** "No prompt phase transitions are observed in both deliquescence and efflorescence measurement modes."

*(7) L342-344, this sentence is hard to understand, please revise it "To have a clear observation for size dependence of the hygroscopic growth factor of D-glucose aerosol nanoparticles with diameters down to 6 nm, Fig. 8b shows the change in the hygroscopic growth factor of D-glucose aerosol nanoparticles with diameters from 100 down to 6 nm."*

**Response:** Thank you for the comments. We have revised in the following sentence and now they read as:

**Line 342-345, we revised:** "Figure 8b further shows the clear change in the hygroscopic growth factor of D-glucose aerosol nanoparticles with diameters from 100 down to 6 nm at 87 % RH."

*(8) L382, has been.*

**Response:** Many thanks. We have revised in the following sentence and now they read as:

**Line 382-383, we revised:** "Thus, nanosize effect on these thermodynamic properties has been taken into account the models and theories (Cheng et al., 2015)."

---

## Author Comment (AC2)

*Response to comments by anonymous referee #2:*

*The manuscript by Lei et al. reported the HTDMA data for levoglucosan and D-glucose particles for the size range of 6 ~ 100 nm. Both of these compounds are atmospherically important chemical species. The data for 100 nm particles agreed well with literature. In the case of levoglucosan particles, measurement of hygroscopicity for particles smaller than 15 nm was challenging due to evaporation. The issue was not observed for D-glucose. In the case of D-glucose particles, a significant size-dependence in hygroscopic growth was observed only for the high RH (RH > 80%) region. The data of the study will serve as a reference for future studies on hygroscopicity of atmospheric nanoparticles. The topic is obviously a part of the scope of the research community. The following points should be considered prior to the acceptance of the manuscript.*

**Response:** We are grateful to referee #2 for the comments and the constructive suggestions. We address in the following the comments and suggestions by referee #2 and provide improvements based on these clarify the questioned issues in the revised manuscript in red. We repeat the specific points raised by the reviewer in italic font in bule, followed by our response. The lines mentioned are with respect to the Atmospheric Chemistry and Physics Discussions (ACPD) version.

*Major comments:*

*(1) The authors state that the UNIFAC model does not provide highly accurate results for organic compounds which contains many polar functional groups (e.g., the description starts from L354). However, the discussion is not well supported by references. Thus, it is difficult for a reader to evaluate the level of uncertainty. As D-glucose is an important chemical compound in various research areas, a numerous number of studies should have been conducted for investigating its interaction with water. It would be great if the authors could provide further detailed discussion using these references to support their argument.*

**Response:** Thanks for the comments. According to the previous studies (Fredenslund et al., 1975; Hansen et al., 1991; Fredenslund and Sørensen, 1994; Mochida and Kawamura, 2004; Lei et al., 2014, 2018), the UNIFAC model was not recommended to predict the hygroscopic behavior of organics whose polar functional groups are separated by less than three or four carbon. For example, Mochida and Kawamura (2004) reported a large disagreement between the UNIFAC model prediction and hygroscopic measurements of levoglucosan and D-glucose containing OH groups

in series, indicating the UNIFAC model without consideration intramolecular interaction. We have removed the following sentence as it is discussed in line 361-364 "Note that, E-AIM (standard UNIFAC) model prediction is optimized for organic compounds with lesser polar groups in series, i.e., intramolecular interaction, such as hydrogen bond between polar 356 groups, may result in model prediction inaccuracy."

The hygroscopic properties of D-glucose aerosol particles are investigated by many groups (Mochida and Kawamura. 2004; Chan and Chan, 2005; Suda and Petters, 2013; Estillore et al., 2017; Mikhailov and Vlasenko, 2020). We have added more discussion as follows:

**Line 357-366, we added references and revised:** "Figure 9a and Fig. S3 show that the measured growth factors of 100-nm D-glucose nanoparticles are lower than predicted growth factors from E-AIM (standard UNIFAC) model, especially at RH below 85 %. Also, E-AIM (standard UNIFAC) model could predict well the measured hygroscopic growth factor of 6-nm D-glucose aerosol nanoparticles at RH above 40 % shown in Fig. 9a and Fig. S3. The possible reason for discrepancies between E-AIM (standard UNIFAC) model and measurements is inaccurate thermodynamic parameters (e.g., water activity, surface tension) estimated by the E-AIM (standard UNIFAC) model without consideration intramolecular interaction (Fredenslund et al., 1975; Hansen et al., 1991; Fredenslund and Sørensen, 1994; Mochida and Kawamura, 2004). D-glucose contains five OH groups in series, hydrogen bond could potentially exist and affects the E-AIM (standard UNIFAC) model-measurement agreement for D-glucose aerosol nanoparticles system (Mochida and Kawamura, 2004; Lei et al., 2014, 2018)."

**Line 325-328, we added references and revised:** "Also, measured hygroscopic growth factors of 100-nm D-glucose are consistent with results from previous studies (Mochida and Kawamura. 2004; Chan and Chan, 2005; Suda and Petters, 2013; Estillore et al., 2017; Mikhailov and Vlasenko, 2020). For example, Mikhailov and Vlasenko, (2020) investigated the hygroscopic behavior of 100-nm D-glucose aerosol particles using a HHTDMA in hydration, dehydration, and restructuring modes of operation, respectively. A clear morphology effect on the hygroscopicity of D-glucose aerosol particles is observed in the RH range from 2 % to 96 % RH."

*(2) It is interesting that the authors only observed the size-dependence in growth factor of D-glucose for the range of RH > 80%. There seems to be no size dependences for the lower RH region. To the best of the reviewer's knowledge, it is not a well-known phenomenon. Do the authors have any potential hypothesis for explaining it? I believe that it will be beneficial for future studies if the author could add some ideas that can explain the observation to the manuscript.*

**Response:** Thanks for the comments. Our hypothesis is that the distinct behaviors between high RH and low RH region can be attributed to the distinct size effect on the hygroscopic growth and adsorption, i.e., the growth factor shows size dependence only in the regime of hygroscopic growth (RH > 80%), and not in the regime of water adsorption (RH < 80%). We have included this hypothesis in the revised manuscript.

**Related additions and changes included in the revised manuscript:**

**Line 340-342, we added references and revised:** "There is no evident difference in hygroscopic growth factors of D-glucose nanoparticles at RH below 80 % in size range from 6 to 100 nm. The reason that the growth factor shows size dependence only in the regime of hygroscopic growth (RH > 80%), and not in the regime of water adsorption (RH < 80%) has not been explained before. Our hypothesis is that the distinct behaviors between high RH and low RH region can be attributed to the distinct size effect on the hygroscopic growth and adsorption, i.e., the growth factor shows size dependence only in the regime of hygroscopic growth (RH > 80%), and not in the regime of water adsorption (RH < 80%).

*Minor comments:*

*(1) abstract: The current abstract is a little bit long and redundant. It describes what the authors have done, but it does not tell the key conclusions of the manuscript well. I suggest the authors to revise it.*

**Response:** Thank you for the comments. We have revised the whole abstract section, and now they read as:

**Line 13-41: Abstract:** The interaction between water vapor and aerosol nanoparticles is important in atmospheric processes. Hygroscopicity of sub-10 nm organic nanoparticles and their concentration-dependent thermodynamic properties (e.g., water activity) in the highly supersaturated concentration range are, however, scarcely available. Here we investigate the size dependence of hygroscopicity of organics (i.e., levoglucosan, D-glucose) in dry particle diameter down to 6 nm using a nano-hygroscopicity tandem differential mobility analyzer (nano-HTDMA). Our results show that there is only a weak size dependent hygroscopic growth of both levoglucosan and D-glucose nanoparticles with diameters down to 20 nm. In the diameter range smaller than 20 nm (down to 6 nm), we observed a strong size-dependent hygroscopic growth for D-glucose nanoparticles. The hygroscopic growth factors cannot be determined for levoglucosan below 20 nm due to its evaporation. In addition, we compare hygroscopicity measurements for levoglucosan

and D-glucose nanoparticles with the E-AIM (standard UNIFAC), the ideal solution theory, and DKA predictions, respectively. The ideal solution theory describes well the measured hygroscopic growth factors of levoglucosan with diameters down to 20 nm and D-glucose nanoparticles with diameters higher than 60 nm, respectively, while the E-AIM (standard UNIFAC) model can successfully predict the growth factors of D-glucose nanoparticles with diameters from 100 down to 6 nm at RH above 88-40 % (e.g., at RH above 88 % for 100 nm D-glucose, at RH above 40 % for 6 nm D-glucose). The use of the DKA method leads to a good agreement with measured hygroscopic growth factors of D-glucose aerosol nanoparticles with diameters from 100 down to 6 nm. Predicted water activity for these aqueous organic solutions (i.e., levoglucosan, D-glucose) from different parameterization methods agrees well with observations in the low solute concentration range ($< 20$ mol kg$^{-1}$), and start to deviate from observations in the high solute concentration ($> 20$ mol kg$^{-1}$)."

*(2) Mochida and Kawamura (2004) are cited for three times for describing the atmospheric significance of levoglucosan and D-glucose. However, the paper by Mochida and Kawamura (2004) focused on hygroscopicity measurement of levoglucosan and other important marker compounds for biomass burning. Other references that directly supports the corresponding descriptions need to be provided.*

**Response:** Thank you for your comments. There are several references supporting the descriptions. For example, Bhattarai et al. (2019) reported the chemical composition of the WSOC fraction of particles generated biomass burning. On the basis of chemical structure, neutral compounds mainly consist of sugar-like compounds such as levoglucosan and D-glucose, which are the most abundant oxygenated products of the pyrolysis of cellulose.

**We have added these corresponding references into the revised manuscript:**

**Line 58-64, we added references and revised:** "Also, levoglucosan is typically the most abundant species in wood burning aerosols, which contributes substantially (16.6–30.9% by mass) to the total organics in PM$_{2.5}$ (Mochida and Kawamura, 2004; Bhattarai et al., 2019). D-glucose, a hydrolysis product of cellulose and levoglucosan, is a major pyrolysis product of wood (Mochida and Kawamura, 2004; Bhattarai et al., 2019; Mikhailov and Vlasenko., 2020). Levoglucosan and D-glucose substances may be representative in reproducing the hygroscopic behavior of the real biomass burning aerosol particles (Bhandari and Bareyre. 2003; Mochida and Kawamura, 2004; Chan et al., 2005; Koehler et al., 2006; Peng et al., 2010; Mikhailov and Vlasenko., 2020)."

*(3) I agree that levoglucosan and D-glucose are important chemical species for biomass burning particles. However, they would probably be one of the most polar chemical species among constituents of biomass burning particles. It is not clear to me if they can be considered as representative compounds for hygroscopicity.*

**Response:** Thank you for your comments.

**We have revised** "Levoglucosan and D-glucose substances may be representative in reproducing the hygroscopic behavior of the real biomass burning aerosol particles." **as** "Hygroscopicity of levoglucosan and D-glucose substances is thus important in reproducing the overall hygroscopic behavior of the real biomass burning aerosol particles."

**Related additions and changes included in the revised manuscript:**

**Line 61-64, we added references and revised:** "Hygroscopicity of levoglucosan and D-glucose substances is thus important in reproducing the overall hygroscopic behavior of the real biomass burning aerosol particles. For example, a small difference in the hygroscopicity parameter ($\kappa$) between measured data of model mixtures including levoglucosan and ammonium sulfate in the laboratory using HTDMA and biomass burning aerosol particles in the field using CCN activity measurement due to the similar O: C ratios of levoglucosan and ammonium sulfate mass fractions used in model mixtures when experimental $\kappa$ data from sub- and supersaturated water vapor conditions are compared (Bhandari and Bareyre. 2003; Mochida and Kawamura, 2004; Chan et al., 2005; Koehler et al., 2006; Peng et al., 2010; Pöhlker et al., 2016; Lei et al., 2018; Mikhailov and Vlasenko., 2020)."

*(4) L123-125 Generally, the issue of multiple charge particles is not a significant concern for small (diameter < 30 nm) particles. Could the authors make the corresponding description to be more detailed?*

**Response:** Thank you for your comments. Yes, reviewer is right, the influence of multiple charges on sub-10 nm particles is expected to be very small. However, the different concentrations are used to make sure that the sizes selected by the nano-DMA1 are always around the peak of the number size distribution of the nanoparticles generated by the electrospray. This is to ensure that we could have as many particles as possible to compensate for the strong loss of very small particles in the whole humidification system.

**Related additions and changes included in the revised manuscript:**

**Line 123-125, we revised:** "Note that the size selected by the nano-DMA1 should be the right part of peak diameter of the number size distribution of the generated nanoparticles, which minimizes

the influence of the multiple charged nanoparticles in hygroscopicity measurements. This is to ensure that we could have as many particles as possible to compensate for the strong loss of very small particles in the whole humidification system."

*(5) L169 The equation (2) looks like an equation for the ideal solution to me. Although the idea of ideal solution can occasionally be applied for the Kohler theory, the Kohler theory is not equivalent as the ideal solution. It would be better to change the name of the corresponding section.*

**Response:** Thank you for your comments. Equation (2) is the Simplified Köhler Theory (SKT) for ideal and diluted solution. We have revised "Köhler" as "SKT" in the whole manuscript.

**Line 26, 106, 166, 233, 234, 235, 314, and 399, we revised "Köhler" as "SKT".**

**Line 692-695 and Line 758-762:**

[Figure]

**Figure 2**. Concentration-dependent water activity ($a_w$) of levoglucosan solution. The KD-derived $a_w$ (KD=Kreidenweis, cyan open square) is compared with observations (red open square), E-AIM (Extend-Aerosol Inorganic Model, black line), and $a_w$ model (SKT, blue line). The light grey shaded areas mark the sub-saturated concentration with respect to bulk solution.

[Figure]

**Figure 6**. Concentration-dependent water activity ($a_w$) of D-glucose solution. The DKA-derived $a_w$ (Differential Köhler Analysis, magenta open square) is compared with observations (red open square), E-AIM (Extend-Aerosol Inorganic Model, black line), $a_w$ model (SKT, blue line), and parameterization model for $a_w$ (KD=Kreidenweis, cyan open square). The light grey shaded areas mark the sub-saturated concentration with respect to bulk solution.

*(6) L235 I wonder what the author's definition of 'diluted aqueous droplet' is. 20 mol kg-1 is highly concentrated. Please clarify.*

**Response:** Thank you for your comments. We have removed the "diluted" and only mentioned the concentration. .

*(7) L237 Please indicate the saturation concentration of levoglucosan in water before mentioning about supersaturation.*

**Response:** Thank you for your comments. According to Zamora et al. (2011), the solubility of levoglucosan is 8.23 mol/kg at 20°C.

We have marked saturation concentration of levoglucosan in the following Fig. 2.

**Related additions and changes included in the revised manuscript:**

[Figure]

**Figure 2**. Concentration-dependent water activity ($a_w$) of levoglucosan solution. The KD-derived $a_w$ (KD=Kreidenweis, cyan open square) is compared with observations (red open square), E-AIM (Extend-Aerosol Inorganic Model, black line), and $a_w$ model (SKT, blue line). The light grey shaded areas mark the sub-saturated concentration with respect to bulk solution.

*(8) L248 (and other places) No deliquescence/efflorescence were observed in the study. It may not be appropriate to call the operation modes of the HTDMA as 'deliquescence/efflorescence modes' under this condition. Hydration/dehydration might be a better expression.*

**Response:** Many thanks. They are the same, and thus we have changed the notion of hydration and dehydration mode to deliquescence and efflorescence modes to avoid confusion.

*(9) Reference: The reference list needs to be carefully checked. There are numerous issues. For instance, I do not believe that the names of the authors of the following paper accurately represented in the current version of the manuscript.*

**Response:** Many thanks. We have carefully checked and revised all reference format, and now they read as:

**Line 458-459:** Chan, M. N. and Chan, C. K.: Mass transfer effects in hygroscopic measurements of aerosol particles, Atmos. Chem. Phys., 60 5, 2703–2712, https://doi.org/10.5194/acp-5-2703-2005, 2005.

**Reference**

Bhattarai, H., Saikawa, E., Wan, X., Zhu, H., Ram, K., Gao, S., Kang, S., Zhang, Q., Zhang, Y., Wu, G., Wang, X., Kawamura, K., Fu, P., and Cong, Z.: Levoglucosan as a tracer of biomass burning: Recent progress and perspectives, Atmos. Res., 220, 20–33, 2019.

Biskos, G., Malinowski, A., Russell, L. M., Buseck, P. R., and Martin, S. T.: Nanosize Effect on the Deliquescence and the Efflorescence of Sodium Chloride Particles, Aerosol Science and Technology, 40, 97-106, 2006a.

Biskos, G., Paulsen, D., Russell, L. M., Buseck, P. R., and Martin, S. T.: Prompt deliquescence and efflorescence of aerosol nanoparticles, Atmospheric Chemistry and Physics, 6, 4633-4642, 2006b.

Biskos, G., Russell, L. M., Buseck, P. R., and Martin, S. T.: Nanosize effect on the hygroscopic growth factor of aerosol particles, Geophysical Research Letters, 33, 2007.

Fredenslund, A., and J. M. Sørensen (1994), Group contribution estimation methods, in Models for Thermodynamic and Phase Equilibria Calculations, edited by S. I. Sandler, pp. 287–361, Marcel Dekker, New York.

Giamarelou, M., Smith, M., Papapanagiotou, E., Martin, S. T., and Biskos, G.: Hygroscopic properties of potassium-halide nanoparticles, Aerosol Science and Technology, 52, 536-545, 2018.

Hämeri, K., Laaksonen, A., Väkevä, M., and Suni, T.: Hygroscopic growth of ultrafine sodium chloride particles, Journal of Geophysical Research: Atmospheres, 106, 20749-20757, 2001.

Hämeri, K., Väkevä, M., Hansson, H.-C., and Laaksonen, A.: Hygroscopic growth of ultrafine ammonium sulfate aerosol measured using an ultrafine tandem differential mobility analyzer, Journal of Geophysical Research: Atmospheres, 105, 22231-22242, 2000.

Martin, S. T. Phase transitions of aqueous atmospheric particles.Chem. Rev. 100,3403–3454 (2000).

Mochida, M. and Kawamura, K.: Hygroscopic properties of levoglucosan and related organic compounds characteristic to biomass burning aerosol particles, J. Geophys. Res., 109, D21202, https://doi.org/10.1029/2004JD004962, 2004.

Lei, T., Zuend, A., Cheng, Y., Su, H., Wang,W., and Ge, M.: Hygroscopicity of organic surrogate compounds from biomass burning and their effect on the efflorescence of ammonium sulfate in

mixed aerosol particles, Atmos. Chem. Phys., 18, 1045–1064, https://doi.org/10.5194/acp-18-1045-2018, 2018

Pöhlker, M. L., Pöhlker, C., Ditas, F., Klimach, T., Hrabe de Angelis, I., Araújo, A., Brito, J., Carbone, S., Cheng, Y., Chi, X., Ditz, 105 R., Gunthe, S. S., Kesselmeier, J., Könemann, T., Lavriˇc, J. V., Martin, S. T., Mikhailov, E., Moran-Zuloaga, D., Rose, D., Saturno, J., Su, H., Thalman, R., Walter, D., Wang, J., Wolff, S., Barbosa, H. M. J., Artaxo, P., Andreae, M. O., and Pöschl, U.: Longterm observations of cloud condensation nuclei in the Amazon 110 rain forest – Part 1: Aerosol size distribution, hygroscopicity, and new model parametrizations for CCN prediction, Atmos. Chem. Phys., 16, 15709–15740, https://doi.org/10.5194/acp-16-157092016, 2016

Romakkaniemi, S., Hämeri, K., Väkevä, M., and Laaksonen, A.: Adsorption of Water on 8−15 nm NaCl and $(NH_4)_2SO_4$ Aerosols Measured Using an Ultrafine Tandem Differential Mobility Analyzer, The Journal of Physical Chemistry A, 105, 8183-8188, 2001.
Zamora, I., Tabazadeh, A., Golden, D., and Jacobson, M.: Hygroscopic growth of common organic aerosol solutes, including humic substances, as derived from water activity measurements, Journal of Geophysical Research (Atmospheres), 116, 23207, 10.1029/2011JD016067, 2011.

---

## Author Response (AR2)

*Response to comments by referee#1:*

*The authors basically answered all the reviewer's concerns. However, there are some issues or minor concerns that the authors should be aware of:*

**Response:** We would like to thank the referee for the constructive comments and suggestions that helped us improve our manuscript. We have implemented changes based on these comments in the revised manuscript. Please find our point-by-point response below. We repeat the specific points raised by the reviewer in italic font, followed by our response. The pages numbers and lines mentioned are with respect to the acp-2022-544-manuscript-version2. PDF that uploaded on 16.02.2023.

*1. A thoroughly check of the manuscript is needed since there are still lot of typos, ill-sentences in the revision.*

**Response:** We have checked and revised all typos and ill-sentences throughout the manuscript.

*1) Line 51: quantification of the biomass burning*

**Response:** Many thanks. We have revised this sentence and now they read as:

**Page 3 line 48-52:** "Levoglucosan aerosol nanoparticles have attracted increasing interest in recent years (Simoneit et al., 1999; Mochida and Kawamura, 2004; Mikhailov et al., 2009; Elias et al., 2010; Lei et al., 2014, 2018; Bhattarai et al., 2019) and is considered as an ideal tracer for characterization and quantification of the biomass burning (Fraser and Lakshmanan, 2000)."

*2) L95-98, in the sentence, what is the role of the where clause? It is hard to understand.*

**Response:** Many thanks. We have revised these sentences and now they read as:

**Page 5 line 93-98:** "Thermodynamic models rely on the concentration-dependent thermodynamic data (such as water activity, liquid-vapor interfacial energy), which are often derived from measurements of large droplet and/or bulk solution (Tang and Munkelwitz, 1994; Tang 1996; Pruppacher and Klett, 1997; Clegg et al., 1998). Nanodroplets can become more highly supersaturated and thus reaching higher solute concentration compared to bulk solution, which makes it difficult for models to predict its hygroscopicity."

*3) L102, understand new particle formation*

**Response:** Many thanks. We have revised this sentence and now they read as:

**Page 5 line 101-103:** "This will further help us to understand the new particle formation, transportation, and their interactions with water molecules."

*4) L61, HTDMA appears in the main text for the first time, and full name should be spelled.*

**Response:** Many thanks. We have provided the full name of HTDMA, Hygroscopicity Tandem Differential Mobility Analyzer.

*5) L124, high solution concentrations*

**Response:** Many thanks. We have revised in the following sentence and now they read as:

**Page 6 line 124-127:** "In order to avoid blocking the 25-μm capillary tube in the electrospray with high concentration solution, the aerosol nanoparticles with diameters of 60-100 and 20 nm are generated by an atomizer with a 0.05 and 0.01 wt % organic solution (i.e., levoglucosan and D-glucose), respectively."

*6) L237, it is apparently an ill-sentence, this is not a dependent sentence, and are should be replaced with of*

**Response:** Many thanks. We have revised in the following sentence and now they read as:

**Page 11-12 line 235-239:** "For example, the hygroscopic growth factors of levoglucosan nanoparticles at 80 % RH, 87 % RH are 1.16, 1.23, respectively, in the deliquescence mode, very close to the corresponding values in the efflorescence mode at the same RH (shown in Fig. S1), suggesting that growing and shrinking of particles are in equilibrium."

*7) I think it is ok to say 20 nm particles instead of 20-nm particles, check throughout the whole text*
**Response**: Many thanks. We have revised in the whole manuscript and now they read as:

**Page 12 line 253-254:** "E.g., a slight difference in hygroscopic growth factor between 100 and 20 nm levoglucosan nanoparticles is ~0.02 at 88 % RH."

*8) L261, with diameters of 100, 60*

**Response:** Many thanks. We have revised in the whole manuscript and now they read as:

**Page 12-13 line 260-263:** "For example, as shown in Fig. 4a, for levoglucosan nanoparticles with diameters of 100, 60, and 20 nm, the thermodynamic equilibrium model (E-AIM (standard UNIFAC)) shows a weak size dependence of the growth factors at low RH but a strong size dependence at RH above 70 %."

*9) L286, how can you image a particle with 0 nm diameter?*

**Response:** Many thanks. We have calculated the saturation ratio of levoglucosan particles against droplet diameters from 1 to 100 nm and revised in the following Fig. 5c and sentence and now they read as:

**Page 14 line 285-286:** "Figure 5c shows the vapor saturation ratio of levoglucosan as nanodroplet diameter increases from 1 to 100 nm."

[Figure]

*10) L290, why use thus here, there is no cause relation according to the context*

**Response:** Many thanks. We have deleted "thus" in the sentence.

*11) L298, evaporation is not countable*

**Response:** Many thanks. We have revised in the following sentences and now they read as:

**Page 14 line 297-298:** "Therefore, there is the obvious partial levoglucosan evaporation from DMA1 to DMA2 within several seconds."

*12) L324, where clause, there is nowhere that can be referred to.*

**Response**: Many thanks. We have revised in the following sentence and now they read as:

**Page 14 line 322-324**: "Estillore et al. (2017) observed a slightly amorphous structure of D-glucose particles under ambient conditions using an atomic force microscopy and D-glucose particles grow through gradual water uptake."

*13) L363, an awkward sentence, I think it should be rewritten*

**Response:** Many thanks. We have revised in the following sentence and now they read as:

**Page 17 line 363-364:** "The ideal solution theory is used to predict the hygroscopic curve of D-glucose nanoparticles with diameters of 6-100 nm, shown in Fig. 9b and Fig. S3."

*2. L337-340, I still don't get it where RH matters here and why it is 80%?*

**Response:** Many thanks. As shown in the following Fig.8a, at about 80 % RH, we observed that an obvious difference in hygroscopic growth factor of D-glucose nanoparticles in the size range from 6 to 100 nm, while a small difference in the hygroscopic growth factor is observed at RH below about 80 %. The reason for the obvious difference in water absorption at high RH ($\geqslant \sim$80 % RH) still needs to be investigated.

[Figure]

**Figure 8: (a)** Hygroscopic diameter growth factor ($G_f$) of D-glucose nanoparticles with dry diameters of 100 nm (red square), 60 nm (blue square), 20 nm (cyan square), 15 nm (green square), 10 nm (pink square), 8 nm (royal square), and 6 nm (black square), respectively.

*3. L370-371, what do you mean "the unfavorable assumption of ideal solution theory"?*

**Response:** Many thanks. For ideal solution, water activity of liquid droplets can be simply estimated from the mole fraction of water. With from 20 down to 6 nm, D-glucose nanodroplets can be highly supersaturated, and the water activity is not equal to mole fraction of water. Thus, with the assumption of idea solution, the model failed to predict the observed growth factors of 6-nm D-glucose nanoparticles at RH above 30 %.

We have revised this sentence and now they read as:

**Page 17 line 370-372**: "For ideal solution, water activity of liquid droplets can be simply estimated from the mole fraction of water. With from 20 down to 6 nm, D-glucose nanodroplets can be highly supersaturated, and the water activity is not equal to mole fraction of water. Thus, with the assumption of idea solution, the model failed to predict the observed growth factors of 6-nm D-glucose nanoparticles at RH above 30 %."